# Mapping Outburst Floods Using a Collaborative Learning Method Based on Temporally Dense Optical and SAR Data: A Case Study with the Baige Landslide Dam on the Jinsha River, Tibet

**Zhongkang Yang** [1,2]**, Jinbing Wei** [1,2,*]**, Jianhui Deng** [1,2]**, Yunjian Gao** [1,2]**, Siyuan Zhao** [1,2] **and Zhiliang He** [2,3]

1   State Key Laboratory of Hydraulics and Mountain River Engineering, Chengdu 610065, China; yangzhk@stu.scu.edu.cn (Z.Y.); jhdeng@scu.edu.cn (J.D.); gaoyj@stu.scu.edu.cn (Y.G.); zhaosiyuan@scu.edu.cn (S.Z.)
2   College of Water Resource and Hydropower, Sichuan University, Chengdu 610065, China; hezhiliang@swust.edu.cn
3   School of Environment and Resource, Southwest University of Science & Technology, Mianyang 621010, China
*   Correspondence: jbwei@scu.edu.cn

**Abstract:** Outburst floods resulting from giant landslide dams can cause devastating damage to hundreds or thousands of kilometres of a river. Accurate and timely delineation of flood inundated areas is essential for disaster assessment and mitigation. There have been significant advances in flood mapping using remote sensing images in recent years, but little attention has been devoted to outburst flood mapping. The short-duration nature of these events and observation constraints from cloud cover have significantly challenged outburst flood mapping. This study used the outburst flood of the Baige landslide dam on the Jinsha River on 3 November 2018 as an example to propose a new flood mapping method that combines optical images from Sentinel-2, synthetic aperture radar (SAR) images from Sentinel-1 and a Digital Elevation Model (DEM). First, in the cloud-free region, a comparison of four spectral indexes calculated from time series of Sentinel-2 images indicated that the normalized difference vegetation index (NDVI) with the threshold of 0.15 provided the best separation flooded area. Subsequently, in the cloud-covered region, an analysis of dual-polarization RGB false color composites images and backscattering coefficient differences of Sentinel-1 SAR data were found an apparent response to ground roughness's changes caused by the flood. We carried out the flood range prediction model based on the random forest algorithm. Training samples consisted of 13 feature vectors obtained from the Hue-Saturation-Value color space, backscattering coefficient differences/ratio, DEM data, and a label set from the flood range prepared from Sentinel-2 images. Finally, a field investigation and confusion matrix tested the prediction accuracy of the end-of-flood map. The overall accuracy and Kappa coefficient were 92.3%, 0.89 respectively. The full extent of the outburst floods was successfully obtained within five days of its occurrence. The multi-source data merging framework and the massive sample preparation method with SAR images proposed in this paper, provide a practical demonstration for similar machine learning applications using remote sensing.

**Keywords:** outburst flood; Sentinel-1; Sentinel-2; threshold; false color composite images; ground roughness; random forest; multi-source geospatial data fusion

## 1. Introduction

In mountainous areas with rugged terrain and narrow valleys, large-scale landslides often cause river blocking and produce severe explosive floods [1,2]. Due to the ultra-high emissions, high-velocity fluxes and long-transport distances, outburst floods can cause severe harm to people and economic losses along hundreds or thousands of kilometers

along the river [3,4]. However, as these hazards tend to occur in harsh environments, complex terrain, and scarce hydrological observation stations, there are very few recorded data on floods caused by landslide dams [5]. There is an urgent need for developing outburst floods mapping methods and enrich observation data for the scientific understanding of geohazard chains that can result from them.

Remote sensing has become an essential tool for large-scale flood disaster monitoring and mapping. In particular, the launch of the Sentinel satellites series increased the availability of free high-quality remote sensing data [6]. Effective spectral indicators have been identified to distinguish open or stagnant waters such as lakes, rivers and coastlines [7] from non-water surfaces [8–11]. Compared with floods during the rainy season in the Asian monsoon regions, the outburst flood is short-lived. Remote sensing images at the peak of the flood are often challenging to obtain. Furthermore, optical images such as those from Sentinel-2 are always expected in cloudy and rainy weather and adequate cloudless data often cannot be acquired within a sufficiently short time interval after a flood.

Synthetic aperture radar (SAR) systems, which actively transmit radar pulse signals and record satellite echo data, has the advantage of providing observation data through cloud cover. Owing to transmitting and receiving radar signals in different polarization modes, a radar image can obtain a rich polarization scattering matrix to reflect the inherent characteristics of ground objects [12]. For instance, the Sentinel-1 SAR satellite is frequently employed for flood detection [13]. Multi-temporal change detection analysis is considered one of the most promising flood extraction methods [14]. The methods based on SAR change detection include the KI algorithm [15], the generalized Gaussian model [16], the regional growth [17], the HSBA-FLOOD method [18], the Change Detection and Thresholding (CDAT) algorithm [19,20] and the Otsu adapting threshold (Otsu) [21]. Although the SAR imaging system has advantages in making up for the lack of optical image data, there are still challenges in distinguishing floods. Since these algorithms merely use image differences, they often identify isolated pixels in non-flood areas as floods and create a 'salt and pepper' phenomenon that results in the miscalculation of the true flood extent [22,23]. Furthermore, in many application scenarios, especially in complex terrain areas, SAR data's noise interference dramatically limits classification accuracy. These noises may be caused by radar mountain shadows, random speckle noise in data acquisition and signal distortion caused by wind or the presence of proximate buildings [24]. The inclusion of additional auxiliary data in flood mapping observably contributes to reduce these mistakes and optimize the classification performance [25]. As floods prefer to occur in low-lying regions, terrain information derived from DEM, such as sinks and streamlines [26,27], are also used to remove the non-flood prone regions. Nardi et al. [28] present the first global floodplain dataset at 8.33 arcsecond resolution with the Shuttle Radar Topography Mission (SRTM) digital terrain model. Some emerging technologies, such as unmanned aerial vehicles (UAVs) and light detection and ranging (LiDAR), can provide high resolution and accurate DEMs to support flood mapping and depth simulation in small-scale [29].

These latest studies used Sentinel-1 or Sentinel-2 data separately in flood mapping. The combined use of time-intensive Sentinel-1 and Sentinel-2 data to improve classification accuracy and timeliness has seldom been researched [30,31]. A number of studies have demonstrated the potential of combining Sentinel-1 data with Sentinel-2 images to support multi-scale flood extending assessment [32], automated training data selection [33] and an increase in observation density during flooding events [34]. The existing combinatorial modes of optical, SAR data for flood mapping usually put the Sentinel-2 multispectral optical image as the central part. VV/VH polarization's backscattering coefficient obtained by processing Sentinel-1 SAR images is added as a separate variable [35,36]. Machine learning algorithms have remarkable advantages in dealing with such multi-source and massive data in remote sensing applications [37]. Due to higher accuracy and anti-noise interference ability, the random forest (RF) algorithm is superior to other classification algorithms. Its good performance in the classification has been proved with a mass of satellite images data [38]. However, when SAR data play a dominant role in flood mapping, constructing

the variable feature set and preparing massive training samples are the primary difficulties restricting the successful implementation of a machine learning algorithm.

While both optical and SAR images have been widely used for open water delineation and flood mapping, there have been few short-duration explosive flood mapping applications. This study aims at addressing the challenges existing in mapping ephemeral floods and cloud covering restriction, taking the outburst flood caused by the landslide dam on the Jinsha River on 3 November 2018, as an example. A new method of collaborative learning with a dense time series of optical images from Sentinel-2, SAR images from Sentinel-1, and DEM data to map the outburst flood is proposed.

## 2. Study Area

### 2.1. The Outburst Flood Resulting from Baige Landslide Dam Break on 3 November 2018

Two river-blocking landslides occurred around the same location on the Jinsha River's right bank in Baige Village, Jiangda County, Tibet in 2018. (Figure 1).

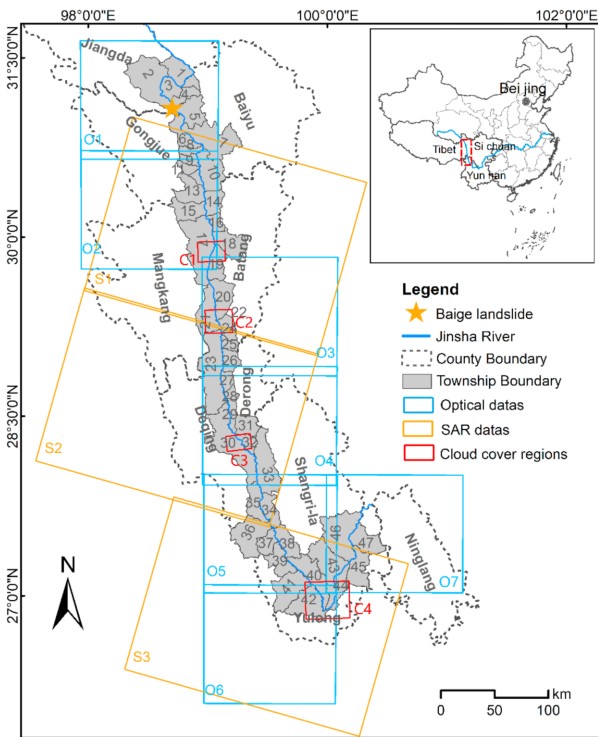

**Figure 1.** Study area location and image range (Number indicates township code. NO 2, 17, 39 and 42 correspond to Bo luo country, Zhu balong country, Ju dian country, and Shi gu country, respectively).

This event provided the valuable opportunity to study the outburst flood resulting from landslide dams [4,39]. The first damming landslide occurred on 11 October 2018 and the barrier lake collapsed naturally two days later. On 3 November 2018, the landslide occurred at the same location once again. The landslide dam height was 73~86 m and the water level increased by 64.04 m, causing a much higher flood threat than the previous one. After constructing an artificial flood discharge channel, the barrier lake with storage of $5.78 \times 10^8$ m$^3$ began flood discharge on 13 November. On 14 November at 13:00, the flood peak arrived at Benzilan Hydrological Station (NO. 30). On 15 November, arrived at Shigu Hydrological Station (NO. 42) at 8:00, arrived at Liyuan Reservoir (NO. 47) at 14:00, and here the flooding process ended. Forty-seven townships were affected, and about 100,000 people were evacuated, resulting in a direct economic loss of 7.43 billion RMB [40]. The study site extends from Jiangda County of Tibet to Ninglang County of Yunnan Province, in the middle reaches of the Jinsha River, China (latitude/longitude: 26°36′43″~31°30′36″, 98°17′44″~100°23′41″) (Figure 1).

## 2.2. Field Investigation

The research team conducted field investigations on the landslide barrier lake and the downstream flooded area from 5 to 16 November and 9 to 19 December 2018, respectively. Our field investigation covered the entire extent of the floods, except for some areas which were restricted to traffic (Figure 2).

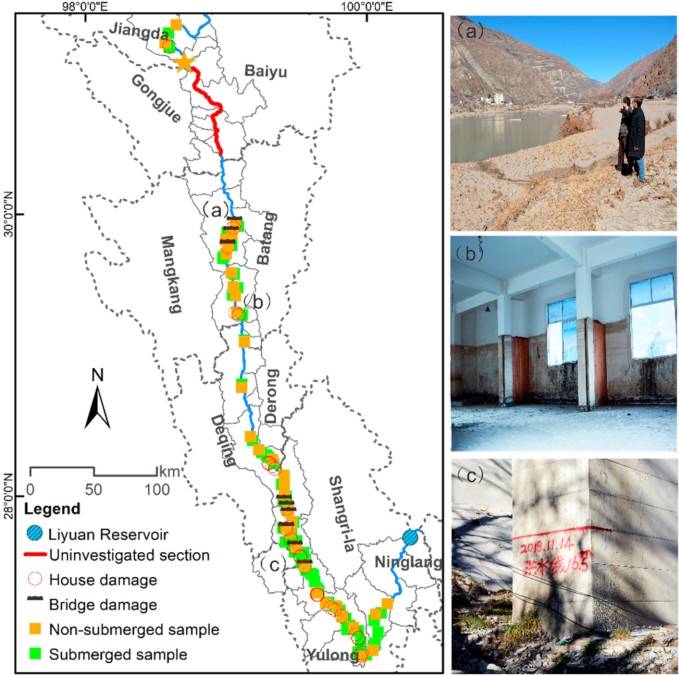

**Figure 2.** Field validation sites. Left: the distribution of validation sites; Right: (**a**–**c**): Scene photos documenting flood damage.

The outburst flood caused substantial erosion and deposition damage to the riversides (Figure 2a, Zhu balong country, NO 17). Nine bridges were destroyed, more than 3000 buildings collapsed, and 3500 hectares of arable land were affected [41]. GPS was employed to record flooded areas (Green squares) and non-flooded areas (Brown yellow square). The flood trace can be surveyed from the wall flood marks (Figure 2b, Ya lang country, NO 23) or bridge elevations recorded (Figure 2c, Ta cheng country, NO 36). A total of 106 validation polygons were collected to evaluate the accuracy of the flood/non-flood pixel classification.

## 3. Data

### 3.1. Sentinel-2 Dates

We downloaded a series of Sentinel-1 and 2 data freely at the Copernicus open access hub from European Space Agency (https://scihub.copernicus.eu/dhus/#/home) (accessed on 25 November 2018). All available Sentinel-2 optical data before and after the flood were obtained (a total of 11 images). The data were available for 12 November, 14 November, and 19 November 2018. According to the image acquisition times, flood process and cloudiness, the flood extraction application was divided into three scenes. In the first scenes, the image can observe the ongoing flood appearance, which is defined as open flood, such as in the landslide barrier lake (Figure 3a, Date: 12 November 2018) and ongoing flood downstream (Figure 3b, Date: 14 November 2018). In the other two scenes, the flooding process ended on 14 or 15 November, but the earliest image available was for 19 November, which failed to observe the flooding. This is thus defined as end-of-flood, which is in a cloud-free region in the second scene (Figure 3c, Date: 19 November 2018) and in a cloud-covered region in the third scene (Figure 3d, Date: 19 November 2018). The spatial distribution and characteristic parameters of the data are shown in Figure 1 and Table 1.

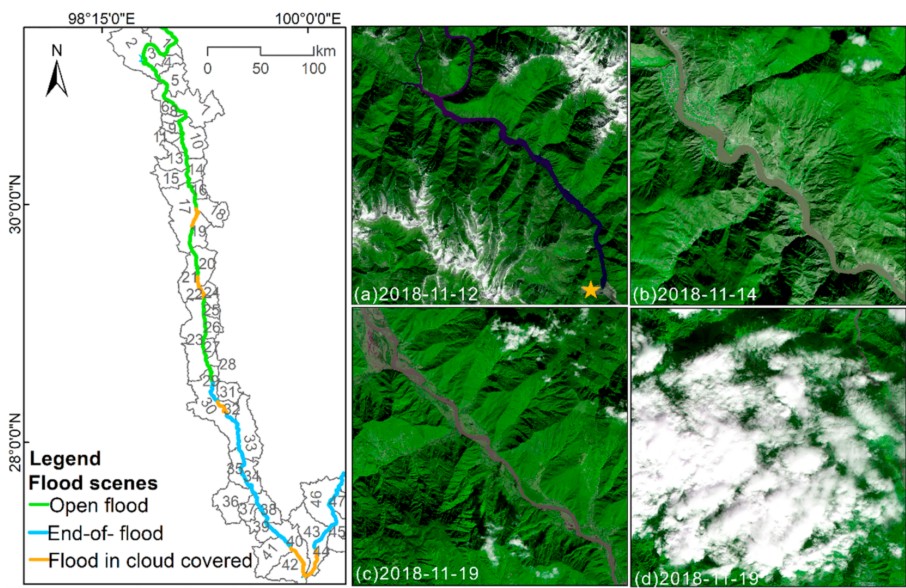

**Figure 3.** Sentinel-2 images examples used with the different outburst flood scenes. Left: Spatial distribution of three flood mapping scenes. Right: Image examples of corresponding scenes, (**a**) Open flood, landslide barrier lake. (**b**) Open flood, ongoing flood. (**c**) End -of-flood in cloudless area. (**d**) End-of-flood in the cloud-covered area.

**Table 1.** Sentinel-2 optical images characteristics used for this study.

| ID | Satellite | Acquisition Time | Flooding Stage | Cloudy Percentage (%) |
|----|-----------|------------------|----------------|----------------------|
| O1 | Sentinel-2B | 20181112T04:10 | barrier lake | 33.44 |
|    | Sentinel-2A | 20181114T04:00 | ongoing | 16.82 |
| O2 | Sentinel-2A | 20181114T04:00 | ongoing | 29.14 |
| O3 | Sentinel-2A | 20181114T04:10 | ongoing | 31.98 |
| O4 | Sentinel-2A | 20181114T04:10 | ongoing | 16.85 |
|    | Sentinel-2B | 20181119T04:00 | ended | 18.88 |
| O5 | Sentinel-2A | 20181109T04:02 | not started | 14.78 |
|    | Sentinel-2B | 20181114T03:59 | not started | 18.3 |
|    | Sentinel-2B | 20181119T04:00 | ended | 38.19 |
| O6 | Sentinel-2B | 20181119T04:00 | ended | 46.17 |
| O7 | Sentinel-2B | 20181119T04:00 | ended | 32.15 |

Major pre-processing procedures for Sentinel-2 images included radiometric calibration, atmospheric correction, and resampling, which were executed on SNAP 8.0.0 (http://step.esa.int/main/download/snap-download/) and Sen2cor plug-in (http://step.esa.int/main/third-party-plugins-2/sen2cor/) (accessed on 25 November 2018). Clouds and shadows detection is essential to limit their interference in optical images. Here, we used for reference to the algorithm of time series the normalized differences index (the combination of blue, the near-infrared (NIR), and the short infrared (SWIR1) channels) from Ludwig et al. [25] to complete the separation of cloud-free and cloud-covered regions. As shown in Figure 1, four main cloud-covered region (C1, C2, C3, and C4) were masked out in the Sentinel-2 optical image along the mainstream of the Jinsha River.

## 3.2. Sentinel-1 Dates

Three Sentinel-1 SAR scenes were downloaded in interference wide mode (IW) to supplement the regions that could not be covered by optical images (Figure 1). The scenes are for 3 November 2018 and 15 November 2018 (acquisition time: 23:20), corresponding to before and end-of-flood scenarios, respectively. Sentinel-1 images with C band obtain

information in VV (vertical transmit radar pulse and vertical receive echo signal) and VH (vertical transmit radar pulse and horizontal receive echo signal) polarization. Sentinel-1 image parameters are shown in Table 2. First, the preprocessing steps of orbit correction, thermal noise removal and border noise removal were executed. Then, speckles resulting from multiple scattering were removed with a Refined Lee filter with a $7 \times 7$ kernel. Finally, we used SRTM1 DEM data to complete the terrain correction and generated the backscatter coefficient of VV/VH in dB.

**Table 2.** Sentinel-2 optical images characteristics used for this study.

| | | | |
|---|---|---|---|
| Acquisition time before flood | 20181103T23:19 | Polarization | VV/VH |
| Acquisition time after flood | 20181115T23:20 | Incidence angle | 39.56° |
| Number of SAR scenes | 6 | Product type | IW-GRD |
| Spatial resolution | 10 m | View geometry | Descending |
| Wavelength | 5.6 cm | Orbit cycle | 154 |

### 3.3. DEM Date and Height Difference

The SRTM1 DEM data (30 m) were used to provide elevation information since the root mean square error with the elevations from field hydrological observation stations was the lowest of all available DEM data (https://earthexplorer.usgs.gov/) (accessed on 25 November 2018). Due to the V-shaped valley terrain, the flooded area was mainly concentrated in the low-lying areas within a height difference of 200 m. Therefore, terrain filters were fabricated to remove highlands that were unlikely to flood. We defined the elevation difference as the difference between each pixel elevation and the lowest point in the catchment unit. The catchment unit was extracted in the ArcGIS 10.2 hydrological analysis modules. Then, the elevation difference data set was calculated in each catchment unit. Finally, a threshold of 200 m was used to obtain the mask for the work area.

We obtained the work area of the "1103" Baige outburst flood, with an area of 587.31 km$^2$, by setting the topographic filter and clipping the images. Based on clouds and shadows detection, the work area was further divided into a cloud-free and a cloud-covered area. The cloud-free area (410.65 km$^2$) was mapped with the Sentinel-2 optical images, identifying 47.14% ongoing flood and 22.78% of end-of-flood. The cloud-covered area accounted for 176.66 km$^2$ and distributed on four sections (C1, C2, C3, and C4) in ongoing flood and end-of-flood regions, which occupied 30.08% of the total work area.

### 4. Methods

Given the challenges existing in mapping the end-of-floods and cloud covering restriction, this study proposed a cooperative learning strategy of optical and SAR images (Figure 4).

(1) In the cloud-free region, we compared four spectral indexes calculated from time series of Sentinel-2 images and found the optimal spectral index and threshold for the end-of flood mapping. (2) In the cloud-covered region, the flood range from the Sentinel-2 image area was taken as prior knowledge, which guided us to explore influential feature variables that could separate flood pixels in the Sentinel-1 images. (3) The RF algorithm's flood prediction framework was structured based on SAR images. The feature set was used: The Hue-Saturation-Value (HSV) color feature of SAR images, the difference or ratio of VV/VH polarization backscattering coefficient, and DEM data are combined to construct 13 factors. Then, the flood pixels obtained from the Sentinel-2 optical images in the cloudless area served as the label set of the training samples, and the pixels of unknown category in the cloud-covered area were used as the prediction samples for the output.

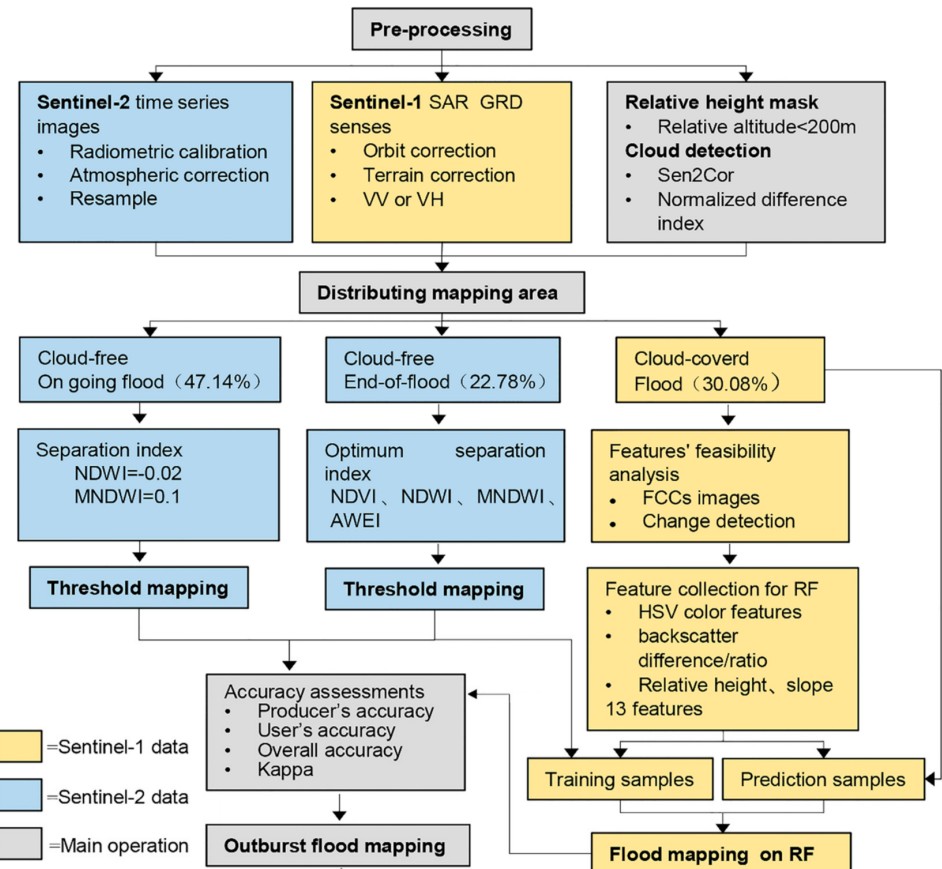

**Figure 4.** The workflow of the proposed method for the outburst floods.

### 4.1. Flood Mapping Based on Sentinel-2 Optical Images in the Cloud-Free Area

4.1.1. Open Water Mapping (Barrier Lake and Ongoing Flood)

For 47.14% of the flooded area in the cloud-free zone, we used both the normalised difference water index (NDWI) and modified normalised difference water index (MNDWI) (Table 3), which all have been shown to be suitable for water recognition in previous studies [8,9]. The indices share the same range of (−1~1), and a general threshold of 0 can be used for separating water from land. Visual analysis indicated that the NDWI threshold was −0.02 and that for MNDWI was 0.1. The flood pixels commonly identified by both indices were used as the final submerged areas.

**Table 3.** Sentinel-2 optical images characteristics used for this study.

| Spectral Indices | Equation | Band in Sentinel-2 | Characteristics | Reference |
|---|---|---|---|---|
| NDVI | $\frac{NIR-R}{NIR+R}$ | $NIR : band8$ $R : band4$ | $-1 \leq NDVI \leq 1$; (0.3~1.0) is vegetation, 0 is rock or bare soil, less than 0 are clouds, water, and snow. | [10] |
| NDWI | $\frac{G-NIR}{G+NIR}$ | $G : band3$ | $-1 \leq NDWI \leq 1$; Positive values are water bodies, negative values are bare soil, vegetation. | [8] |
| MNDWI | $\frac{G-SWIRI}{G+SWIRI}$ | $SWIRI : band11$ | The same features as NDWI. It enhances the contrast between water and buildings. | [9] |
| AWEI$_{nsh}$ | $\frac{4(G-SWIRI)}{0.25NIR+2.75SWIR2}$ | $SWIR2 : band12$ | Further removing shadows are easily confused with water. | [11] |

### 4.1.2. Optimal Spectral Index for End-of-Flood Mapping

Based on the experience of Ludwig et al. [25] and Slagter et al. [42] in wetland classification, we exploited the properties of wet surfaces to strongly absorb radiation in the NIR and SWIR1 bands, compared to dry soil or vegetation reflecting the wavelengths. For 22.78% of the end-of-flood area in the cloud-free zone, we compared the four most promising spectral indices, NDVI (normalized difference vegetation index), NDWI, MNDWI and AWEI$_{nsh}$ (the automated water extraction index, referred to as AWEI), which are all based on combinations of the NIR and SWIR1 bands. The calculation formulas and characteristics are shown in Table 3.

To verify the usefulness of these spectral indexes for end-of-flood mapping, the change values were compared before and after flood occurrence in 5 adjacent days. The Sentinel-2 image area O5 was taken as an example (Figure 5). On 9 and 14 November before flood, the spatial distributions of spectral values of the two images are consistent. While during the following 5-day intervals between 14 and 19 November, the spectral indexes underwent drastic changes. There was a substantial increase in the blue or blue-purple color areas indicate water or hydrous surface, extending about 2 km inland of the riverbank at its widest. Furthermore, NDVI showed the most apparent changes caused by the end-of-flood, followed by NDWI.

To further confirm the optimal spectral index and accuracy of the threshold range, the changes in spectral values in the river section were quantitatively analyzed. Consequently, the representative observation section b was selected (Figure 5 red line b) on account of a canyon topography and a full range of surface features, such as bottomland, bare land, woodland, farmland, and buildings. Distinguishing the end-of-flood is based on the observation before and after flood during a period of 5 days. Each spectral index has a representative spectral interval for water and dry surfaces (Table 3). In comparison, the transition zone from a typical dry surface to water is easily inferred as a humid surface [43,44], which provides critical evidence for identifying traces of flood inundation.

River section b, the flood elevation is 1853 m and the flood range mainly concentrates in 0~0.9 km, located on the right side of the river's bank. Taking the NDVI curve as an example (Figure 6), after the flood transited on 15 November, the value dropped to (−0.1~0.2) on 19 November, which is lower than that of the dry surface (0.3) and slightly higher than that of water (0), indicating the wet status of the end-of-flood pixels. In addition, only NDVI and NDWI can be successfully used to identify flooded pixels at elevations below 1853 m via appropriate thresholds on 19 November. MNDWI and AWEI show "foreign objects with the same spectrum" noise on the left highland (−0.3~−0.8 km). Those spectral values of non-flooded pixels are similar to those of flooded pixels on the right bank, where the MNDWI value intervals are (−0.25~−0.22) and those of AWEI are (−1.25~−1). We found the same false wet surface noises on the other steep mountains oriented southward. Therefore, we explored the use of NDVI and NDWI for end-of-flood mapping, and the potential threshold range was (0.05~0.20) for NDVI and (−0.30~−0.15) for NDWI.

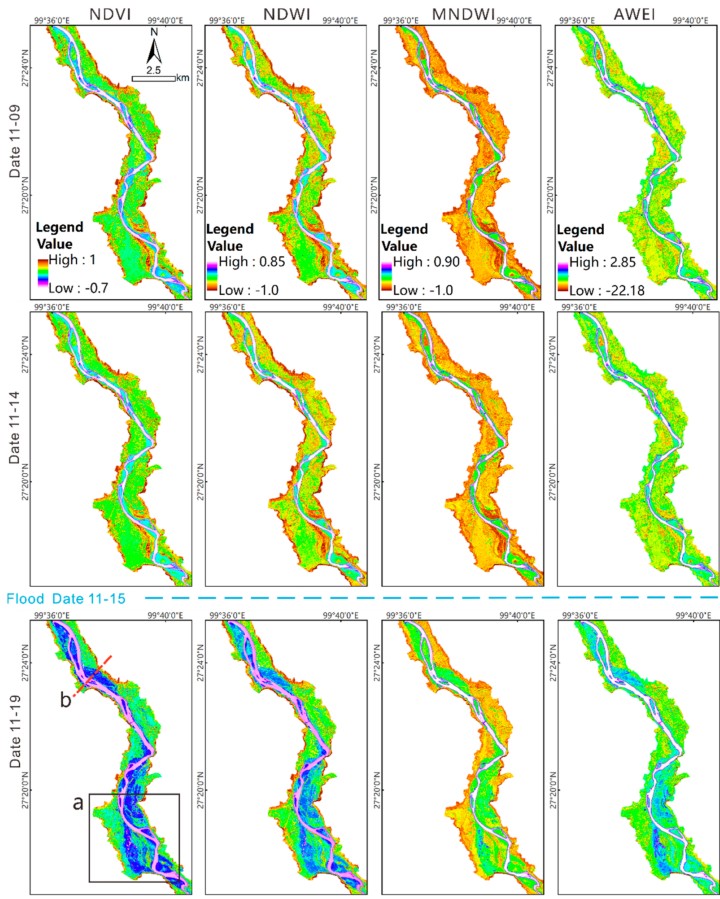

**Figure 5.** The Sentinel-2 four spectral indices. The time series were from 11.09 to 11.19. From left to right, are NDVI, NDWI, MNDWI, AWEI, respectively. From top to bottom are observation dates 11.09, 11.14 and 11.19 in order; The blue dashed line between images on 11.14 and 11.19 indicates the flood occurred date was 11.15. Line b is the river section location in Figure 6 and rectangular a is the test area of Figure 7.

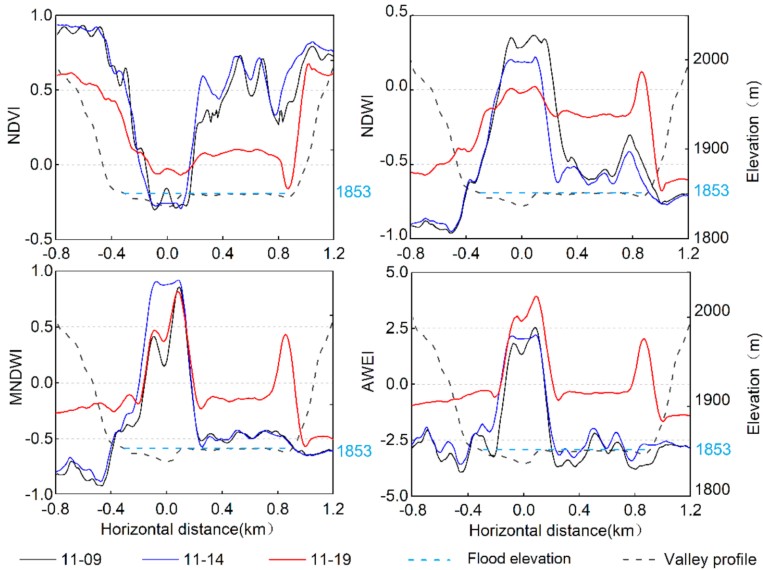

**Figure 6.** Time series of four spectral indices of profile b.

### 4.1.3. Optimal Threshold for End-of-Flood Mapping

Based on the results in the previous section (Section 4.1.2), a test area around Judian Town (No. 39) (Figure 5, rectangular a) was selected. Through artificial visual optimization (Figure 7), we identified the thresholds of NDVI (0.15) and NDWI (−0.19) to be most consistent with the flood scene photos. Particularly, when NDVI < 0.15, the inundation area is remarkably consistent with the actual flood range (Figure 7, A–C). The partition of the non-submerged area was also correct, as can be seen from the highland (Figure 7, F) in the left bottom region and the lowland in the left corner (Figure 7, D).

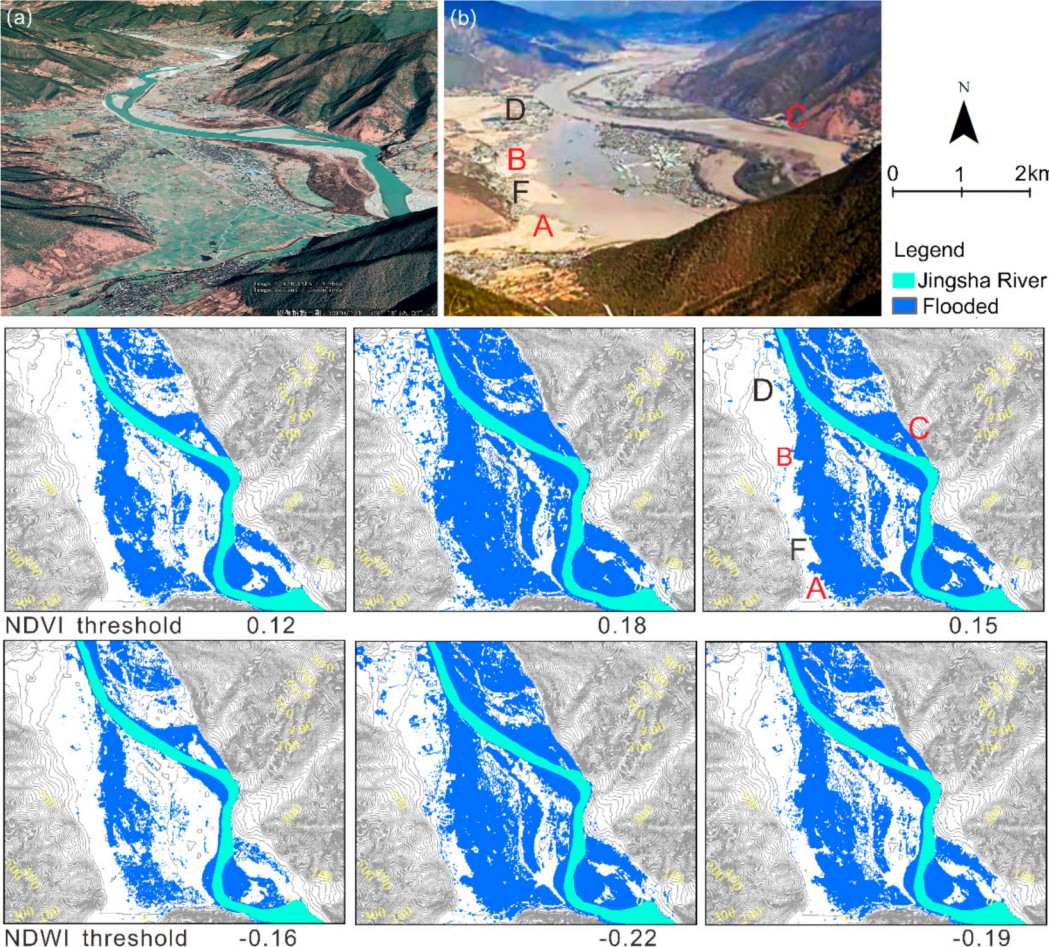

**Figure 7.** Flood mapping experiments obtained from applying NDVI (middle row) and NDWI (last row) to the test site in Ju Dian Town using Sentinel-2 images on 19 November 2018. (**a**) Google 3D view showing of the test site, (**b**) Flood transiting scene photos for 8:00 a.m., 15 November 2018.

### 4.2. Effective Flood Mapping Features Based on Sentinel-1 SAR Images

#### 4.2.1. RGB False Color Composites Images

The method of RGB false color composites (FCCs) can be applied to multi-polarization backscattering intensity to generate a visual color effect just like with optical data [45]. The FCCs of Sentinel-1 SAR image before and after the flood was produced for reference to this background enhancement method. As shown in Figure 8a,b, there was no significant difference between the two grayscale images before (3 November) and after (15 November) the flood. A random combination of VV/VH polarization SAR images on 3 November and 15 November shows unique color features in aqua (Figure 8c), bright green (Figure 8d) and deep red (Figure 8e). The flood inundation region obtained from the Sentinel-2 optical image was taken as a priori knowledge for comparison. When the color channels were

assigned as follows: R: 11.03 VV; G: 11.15 VV; B: 11.15 VV (Figure 8f), the known end-of-flood pixels could be enhanced the visual display.

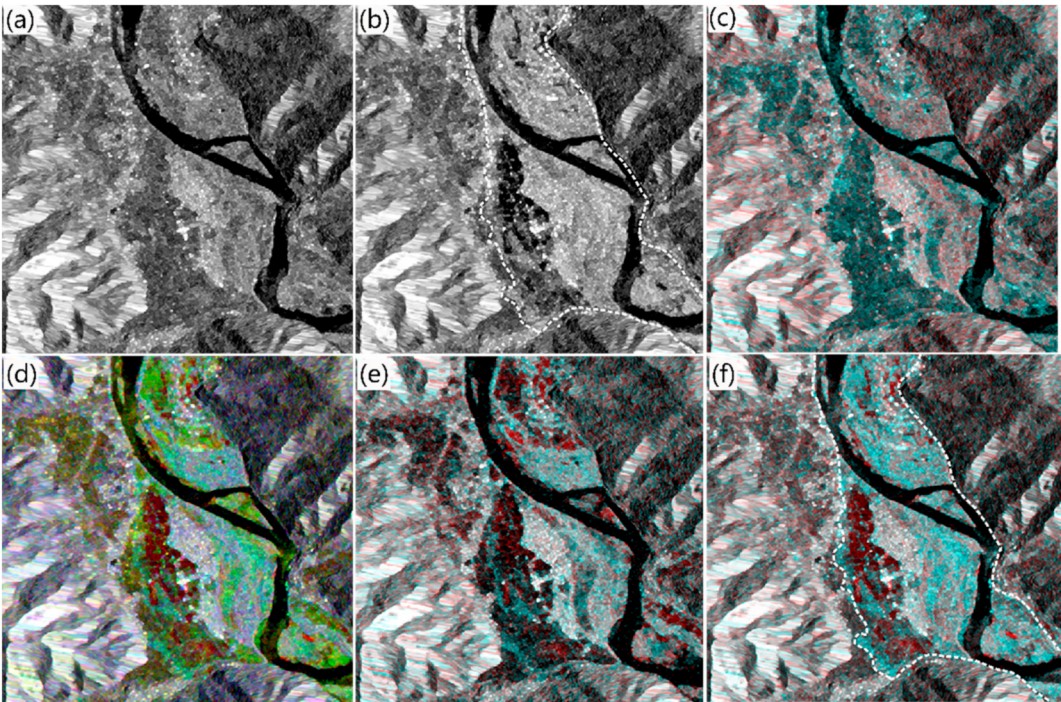

**Figure 8.** Sentinel-1 FCCs images with difference combination of SAR polarization. (**a**) VV SAR image on 2018.11.03; (**b**) VV SAR image on 2018.11.15; (**c**) R: 1103VV, G: 11.15VV, B: 11.15VH; (**d**) R: 1103VH, G: 11.15VV, B: 11.15VV; (**e**) R: 1103VV, G: 11.15VH, B: 11.15VV; (**f**) R: 1103VV, G: 11.15VV, B: 11.15VV. The white dotted line is the inundation range from Sentinel-2 data.

Considering the interference of shadows from steep terrain, the validity of the FCCs images was further tested on the valley sections of C1, C2 and C4. As shown in Figure 9a–c, these inundated areas in the valley continue to be characterized by different aqua green and deep red colors. Superimposing the grayscale back-scattering images before and after flood indicated that the end-of-flood pixels could be distinguished, providing a visual basis for the manual delineation of the end-of-flood in the cloud-covered area.

### 4.2.2. The Backscatter Difference on Sentinel-1 SAR Date

The backscatter coefficient change map of corresponding regions was calculated as the difference between the VV-polarized SAR image of 15 November 2018, and 3 November 2018. When the FFCs images tend to show aqua green, the backscattering change is positive. As shown in Figure 9a–c, these bright aqua green regions show high positive values in the backscatter coefficient difference (above 10) (Figure 9a,c,d). The FFCs images tend to be deep red, and the corresponding backscatter coefficient change is negative. In Figure 9a, the deep red regions in the river bend correspond to high negative values of (below −10) (Figure 9b). Based on these observations, we can infer that the backscatter coefficient difference is an influential variable for outburst flood mapping as well. In despite of complicated bidirectional change, these high value difference regions have a good correspondence with the flood pixels referred to FCCs images.

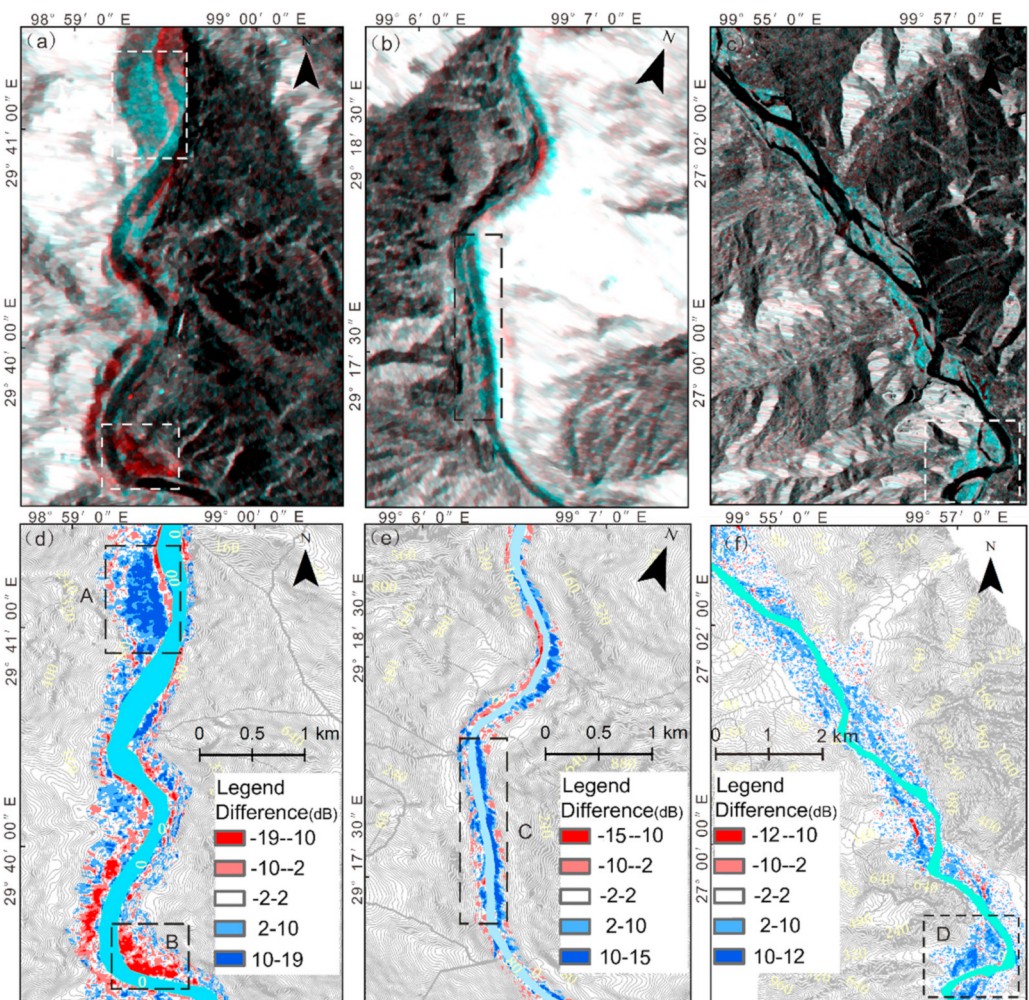

**Figure 9.** Sentinel-1 SAR FCCs images and VV polarized backscattering coefficient difference. (**a**–**c**) are FCCs images from C1, C2 and C4 in Figure 1, respectively. (**d**–**f**) are VV polarized backscatter coefficient differences.

### 4.3. Flood Classification by RF Algorithm on Sentinel-1 Images in the Cloud-Covered Area

#### 4.3.1. RF Algorithm

As an efficient ensemble learning algorithm, RF algorithm has been widely used to classify many multi-source data in satellite images [37,46]. The RF model is composed of many individual decision tree models that each tree fits a data subset sampled independently using bootstrapping. Out of bag data is used to get both variable importance estimations and an internal unbiased classification error as trees are added to the forest, while bagging form of bootstrapping is used to randomly select samples of variable as the training dataset for model calibration [47]. The outburst flood prediction model was divided into three steps: Preparing training data sets, bootstrap samples for decision trees and generation algorithm. In preparing the training data sets, a wide variety of relevant features are necessary to reduce classification errors. To avoid over-fitting problems caused by a data imbalance, the training set should be very large and balance the amount of flood and non-flood samples before bootstrap samples. The optimal parameters of the model are estimated by gradual optimization.

#### 4.3.2. Feasible Features Combination

#### HSV Color Feature Extraction

To increase the color features' effectiveness, the FCCs image was converted to the HSV color space, which transforms the composite into hue, saturation, and value. Hue

represents the position of the spectral color. The hues of red, green, and blue are apart from 120°. Saturation represents the ratio between the selected color saturation and the maximum saturation. The value represents the brightness of the color. The conversion of RGB to HSV is calculated as follows [48]:

$$V = \max(R, G, B) \tag{1}$$

$$S = \frac{\max - \min}{\max} \tag{2}$$

$$H = \begin{cases} 60\frac{G-B}{\max-\min} & R = \max \\ 60\frac{2+B-R}{\max-\min} & G = \max \\ 60\frac{4+R-G}{\max-\min} & B = \max \\ H + 360 & H < 0 \end{cases} \tag{3}$$

where *R G B* correspond to the red, green and blue channel, respectively, in this study, *R* is 11.03 VV polarization; *G* is 11.15 VV polarization; *B* 11.15 VV polarization; max is the maximum of *R*, *G*, *B*; min is the minimum of *R*, *G*, *B*.

Calculation of Backscatter Difference/Ratio Images

The change map of the polarization backscattering coefficient is the primary feature of SAR images. VV polarization has an advantage in distinguishing surface roughness changes, while the VH polarization is more sensitive to scattering changes of specific ground objects [20]. The polarization scattering coefficient difference/ratio vectors are constructed by intersecting VV and VH polarization data to fully extract the change information of the backscatter coefficient. The calculation formula is as follows:

$$g(x,y) = g_{mn}^{N+1}(x,y) - g_{mn}^{N_1}(x,y) \tag{4}$$

$$f(x,y) = \frac{f_{mn}^{N+1}(x,y)}{f_{mn}^{N_1}(x,y)} \tag{5}$$

where $g(x,y)$ and $f(x,y)$ denotes backscatter difference and ratio, respectively. *m* and *n* denote VV polarization and VH polarization, respectively. *x* and *y* denote the rows and columns of pixels in the image. N + 1 denotes the images on 15 November after the flood, and N1 denotes the images on 3 November before the flood.

Low-lying and flat areas are more likely to be flooded. Hence, the elevation difference and slope obtained from DEM data are also a necessary auxiliary dataset. The dataset integrating SAR image color features, scattering differences features and DEM data is shown in Table 4, with 13 variables.

**Table 4.** Feature set for RF algorithm acquired from multi-source data.

| Family | Feature |
|---|---|
| HSV Color space (3) | Hue, Saturation, Value |
| Differences (4) | $VV_{N+1} - VV_{N1}$, $VH_{N+1} - VH_{N1}$, $VV_{N+1} - VH_{N1}$, $VH_{N+1} - VV_{N1}$ |
| Ratio (4) | $VV_{N+1}/VV_{N1}$, $VH_{N+1}/VH_{N1}$, $VV_{N+1}/VH_{N1}$, $VH_{N+1}/VV_{N1}$ |
| DEM (2) | Height difference, Slope |

4.3.3. Training and Prediction Data

Whereas preparing massive training data through manual collection is time-consuming, this paper uses the flood pixels obtained from the Sentinel-2 optical image in the cloudless area as a label set of the training samples (the submerged/non-submerged pixels have values of 1/0, respectively). A total of 1,263,500 training samples and 642,900 prediction

samples were prepared (part as shown in Figure 10a). The ratio of flood to non-flood pixels in the training samples approached 5:5.

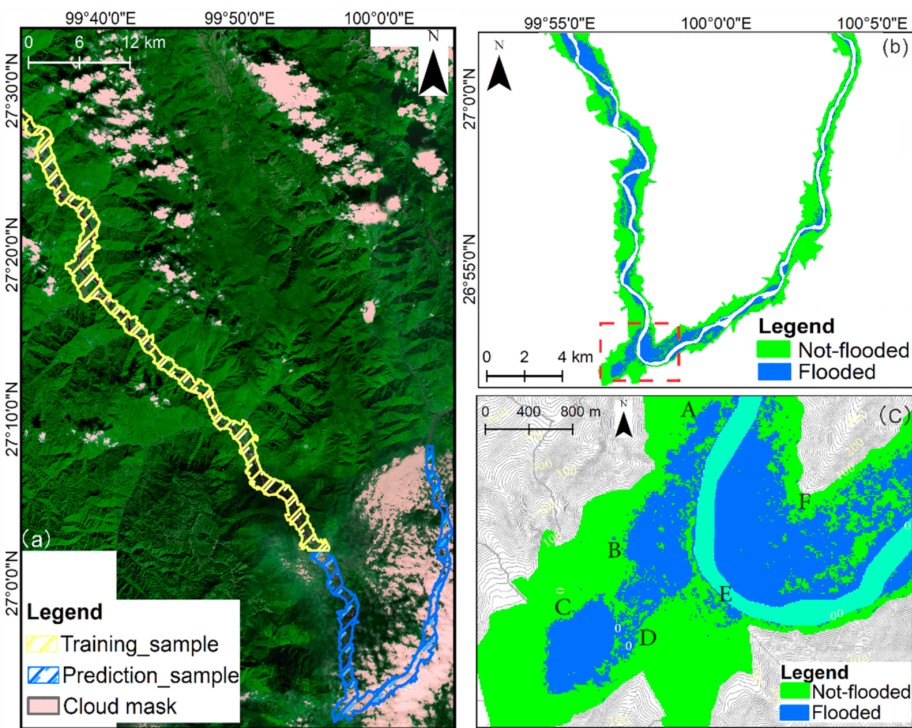

**Figure 10.** RF algorithm flood mapping result. (**a**) Spatial distribution of training samples and prediction samples, the cloud occlusion area C4 (Shigu Town, NO. 42); (**b**) Flood prediction results. (**c**) The detailed mapping results of the red box.

## 5. Results

### 5.1. Prediction Mapping of RF Algorithm on Sentinel-1 Images

We executed the RF algorithm with Python 3.7 pandas and the scikit-learn module. The optimal number of subdecision trees in the RF algorithm external frame was 101, which was generated from the parameter optimization process from 0 to 120. To ensure the efficiency of the operation, the maximum tree depth of each decision tree was set to 7. The optimal parameters were selected to rerun the training set and the generalization error was 98.63%.

The predicted results of the cloud-coverd region C4 were presented in Figure 10b. To further evaluate and validate classification results, we magnify the area within the red box shown in Figure 10c. The RF algorithm partitions the flooded area into pure and compact objects. A distinct and integrated boundary is observed between the inundated and the non-inundated area. The feature points of the inundated boundary are consistent with the flood site photos (Figure 11, A–F). This indicates that the RF algorithm has an excellent performance in the identification of the end-of-flood. For comparison, the flood mapping methods of the (CDAT) algorithm [20] based on VV polarization were also executed. In contrast, the results show widespread 'salt and pepper' and are not properly able to identify the end-of-flood area (Figure 11d).

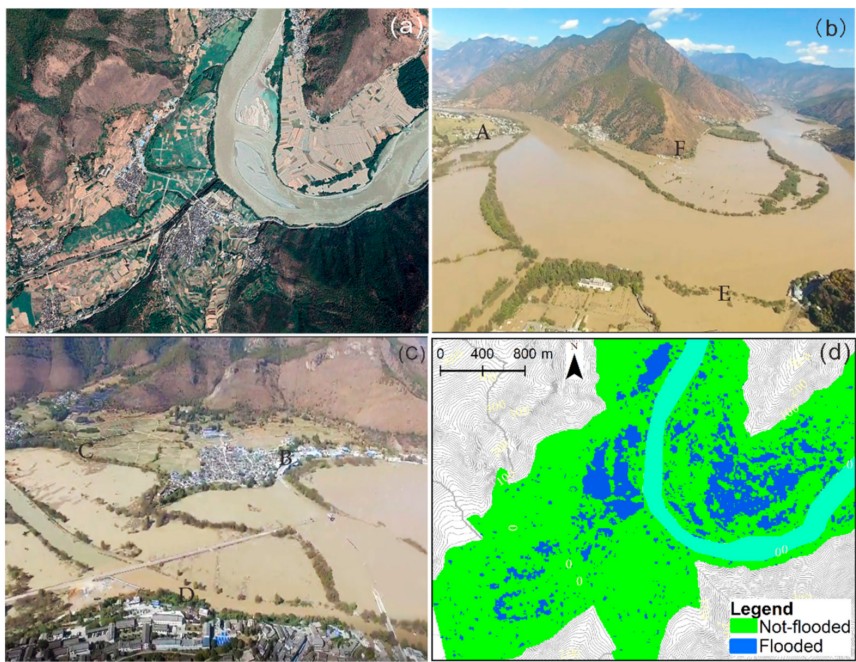

**Figure 11.** The mapping results accuracy evaluation. (**a**) Google Image; (**b**,**c**) Flood transit scene photos. Filming time: 12:00 a.m. of 15 November 2018. (**d**) the classification results of the region using CDAT algorithm.

### 5.2. Flood Mapping Accuracy

The 106 flood verification areas collected in the field were converted into verification points with a resolution of 10 m, and validation analysis using a confusion matrix was constructed. According to user's accuracy (UA), producer's accuracy (PA), overall accuracy (OA) and Kappa coefficient [49], the accuracy of flood mapping based on Sentinel-2 threshold classification and Sentinel-1 RF algorithm were assessed.

The classification accuracy of the Sentinel-2 optical image mapping area is shown in Table 5. The highest flood classification accuracy was obtained at NDVI < 0.15. The overall accuracy and Kappa coefficients are 95% and 0.93, respectively. The UA and PA of flood pixels and non-flood pixels are above 93%.

**Table 5.** Flood mapping with NDVI and NDWI accuracy assessment results on the Sentinel-2 image.

| Classification Method | Threshold | Classification | UA/% | PA/% | OA/% | Kappa |
|---|---|---|---|---|---|---|
| NDVI | 0.16 | Flooded | 92.28 | 96.92 | 94.5 | 0.92 |
| | | Non-flooded | 96.94 | 92.21 | | |
| | 0.15 | Flooded | 96.16 | 93.8 | 95 | 0.93 |
| | | Non-flooded | 93.9 | 96.2 | | |
| NDWI | −0.20 | Flooded | 91.89 | 96.4 | 94 | 0.92 |
| | | Non-flooded | 96.32 | 91.6 | | |
| | −0.19 | Flooded | 95.73 | 93 | 94.4 | 0.92 |
| | | Non-flooded | 93.14 | 95.8 | | |

In the Sentinel-1 SAR image mapping area, we compare firstly the accuracy between RF algorithm with crossed change backscattering variables and the methods merely using VV polarization. The overall accuracy and Kappa coefficient of the flood classification for the RF algorithm are 90.4% and 0.85, respectively (Table 6), which were 12.67, 0.27 higher than the CDAT algorithms, respectively. Furthermore, it was found that the combination of HSV color parameters, SAR backscattering variables, and DEM data achieved the highest classification accuracy. The overall accuracy and Kappa coefficient were 0.19 and 0.04 higher than those using only change backscattering variables.

**Table 6.** Flood mapping accuracy assessment results on the Sentinel-1 SAR date.

| Classification Method | Classification | UA/% | PA/% | OA/% | Kappa |
|---|---|---|---|---|---|
| CDAT VV | Flooded Non-flooded | 76.11 82.2 | 80.35 84.13 | 77.73 | 0.58 |
| RF Changes | Flooded Non-flooded | 92.84 87.67 | 90 90.86 | 90.4 | 0.85 |
| RF Changes + HSV + DEM | Flooded Non-flooded | 93.39 92.21 | 92.2 91.57 | 92.3 | 0.89 |

*5.3. Flood Area Statistics*

Within five days of '1103' Baige landslide dam outburst flood, the quantitative flood inundation area of 47 towns were obtained, and the spatial distribution was shown in Figure 12.

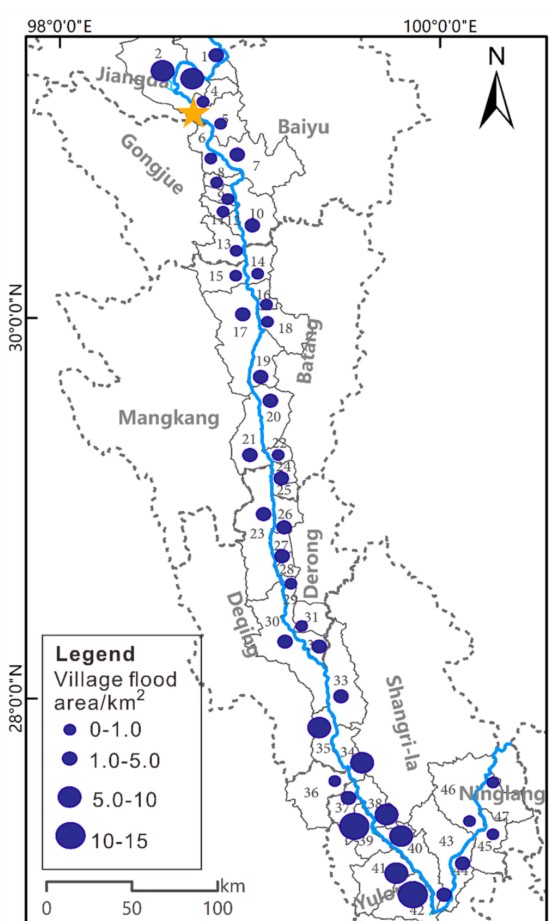

**Figure 12.** The spatial pattern of flooded area about "1103" Baige Landslide outburst flood.

Excluding the original water area (the Jinsha River area was 77.28 km$^2$), the total flooded area amounted to 101.75 km$^2$. In the barrier lake region, four townships were disturbed, and Bolo township (NO 2) was the largest, with an area of 9.86 km$^2$. Downstream of the outburst point, there were more than 25 townships with the flood inundation area of 0~5 km$^2$. The distribution of the larger flood area was closely related to valley topography and mainly concentrated in Jinsha River's broad and low valley. The five largest-affected towns are all located in the downstream Yu long country, Shangri-la country of Yunnan Province, followed by Judian Town (NO 39), Shigu Town (NO 42), Jinjiang Town (NO

40), Shangjiang Town (NO 38) and Liming Town (NO 41). The average submerged area is 10.83 km$^2$, more than 500 km away from the barrier dam burst point.

## 6. Discussion

### 6.1. The Relationship of Backscatter Differences and the Outburst Flood

As a water surface is a specular reflector of radar signals, the echo signal received by the satellite over water is very small. Compared with the stronger land scattering, flooding generates a sizeable negative difference in backscattering coefficient [50]. However, unlike the unidirectional negative change for flood mapping, in this case, there is a complex bidirectional change (both positive and negative). We explored the nature relationship of the bidirectional changes in the backscatter coefficient and the outburst flood by analyzing field investigation. In general, changes in the backscatter coefficient of SAR images are caused by variations in surface roughness, moisture content, and slope [51]. Surface roughness is an especially crucial factor, and the rougher the surface, the stronger the backscatter [52]. Extensive field investigations show that the high-speed sand-gravel outburst flood caused severe erosion and sedimentation along the Jinsha River, leading to the apparent surface structure and backscatter coefficient changes after the flood. The regions analyzed in Figure 9 are examples. As shown in Figure 13A (the corresponding region in Figure 9a), the high-velocity flood scoured the open ground on the left bank, impacting an area of 54,428.7 m$^2$.

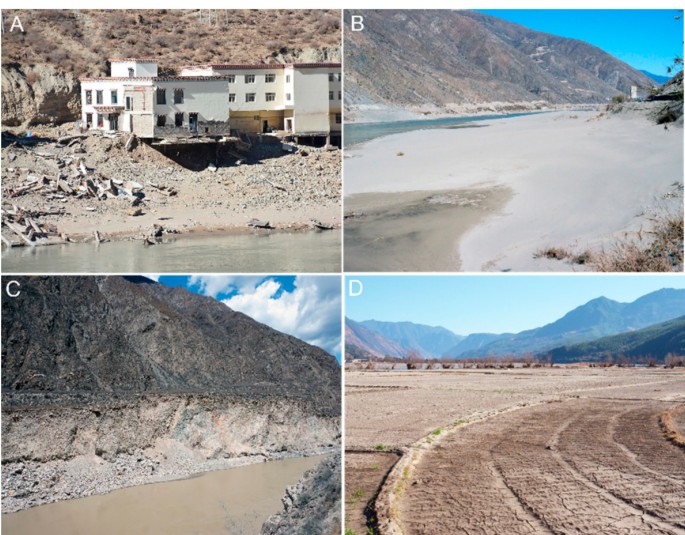

**Figure 13.** Damage photos of scouring and silting after the flood. (**A–D**) correspond to areas in Figure 9, (A–D).

The collapsed buildings and the fragmentation of the bed roughed the surface structure. As a result, the ground backscatter was increased, resulting in a positive difference in backscatter. A similar situation can be observed in Figure 9c, where the roads were destroyed, and their width went from 5.0 m to 2.6 m. The fresh sharp bedrock of the subgrade was exposed and increased the ground backscattering (Figure 13C). Moreover, following hydrodynamic force weaken downstream, the ability of the flood to rough the surface decreases accordingly. Wherefore the difference in Figure 9c is mainly between 2 and 10 and the disturbance of cultivated land is like in Figure 13D. For the negative values of the backscatter coefficient differences (Figure 9b), the flood ended on 14 November. Wash load and suspended deposition subsided at the steep river bend and the river shoal area on the left increased from 31,945 m$^2$ to 38,296 m$^2$. Fine and smooth sediment particles covered the gravel on the beach with a thickness of up to 1.3 m (Figure 13B). The newly emerged smooth surface decreased the intensity of the ground backscatter. Unlike the

change in water content, the surface roughness change is durable, providing the variables needed for distinguishing the end-of-flood from SAR images.

### 6.2. Reliability of Flood Mapping Using a Collaborative Learning Method

Compared with the information obtained from HH polarization in the L band (such from PALSAR), the C band of the Sentinel-1 satellite provides limited change detection ability [53]. The method in this study overcomes these limitations, demonstrating an excellent performance. High accuracy flood ranges from Sentinel-2 image laid a good foundation in collaborative learning for Sentinel-1 flood mapping. The construction of relevant features combination plays a crucial role as well. The weight of the 13 variables were ranked via RF algorithm, as was shown in Figure 14. The variables whose importance exceeded 0.1 were, in order, height difference, hue, slope, $V_{N+1} - VN_1$ and $VV_{N+1} - VH_{N1}$. This shows that the elevation difference, topographic slope and hue parameters of the color features are fundamental to flood pixel prediction. The detailed information indicating flooded areas hidden in the grayscale SAR images was highlighted with FCCs images. Moreover, the color features and the DEM data show more uniform gradient properties and lower variability compared with the backscatter coefficient differences. This combination use can efficiently reduce the speckle noise inherent in SAR multi-polarization scattering images. Consequently, the non-flooded pixel UA was increased from 87.67% (Changes) to 92.21% (Changes + HSV + DEM). In addition to the $V_{N+1} - V_{N1}$ polarization variable, the cross-polarization difference/ratio ($VV_{N+1} - VH_{N1}$, $VV_{N+1}/VH_{N1}$) are sensitive to the change of ground roughness. When the data set combined the crossed VV and VH polarization, both the PA and UA of the flood pixels and non-flood pixels have been improved notably, showing a better monitoring ability than single-polarization data.

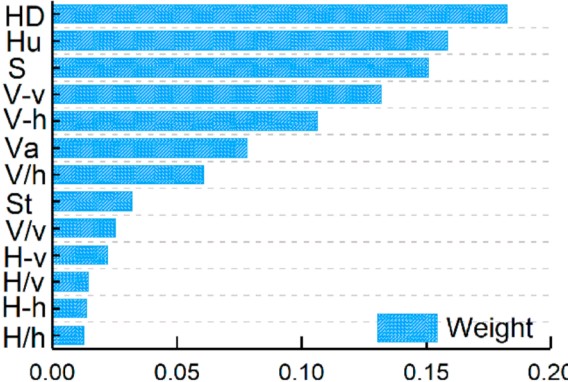

**Figure 14.** Feature importance ranked by the RF algorithm, HD, S is relative height difference and slope, Hu, St, and Va are hue, saturation, value of the HSV Color space; V,v is $VV_{N+1}$ and $VV_{N1}$ and H, h are $VH_{N+1}$ and $VH_{N1}$ respectively.

As this was the first application of short-duration flood mapping, the preprocessing, spectral analysis, and machine learning of the original images were performed on SNAP, ArcGIS, ENVI and Python platforms, which required rereading the data several times, lowering the processing speed. The use of a cloud platform such as Google Earth Engine can improve processing efficiency in future work.

## 7. Conclusions

Outburst floods caused by large-scale landslides create hazards that are often more severe than the landslides themselves. Given the challenges existing in mapping ephemeral outburst floods in cloud-covered areas, the RF algorithm's flood mapping framework was based on SAR images. A combination of features with 13 variables integrating SAR FCCs image, backscatter differences and DEM data was created and a massive label set of training samples was acquired from Sentinel-2 optical images, providing a practical demonstration for similar machine learning applications using SAR images. The combination of the

Hue-Saturation-Value color features and DEM can efficiently reduce the speckle noise inherent in SAR multi-polarization scattering images. Despite the practical barriers owing to image acquisition, format conversion and computational burden, field investigations and accuracy assessments have demonstrated the excellent performance of this method. This study presented the first mapping application of short-duration floods derived from landslide dam breaking using remote sensing images, which can be used to accelerate the evaluation of flood loss and emergency response in landslide hazard chains.

**Author Contributions:** Z.Y.: writing—original draft preparation; J.W.: writing—review and editing and funding acquisition; J.D.: Investigation, Resources, and Project administration; Y.G. and S.Z.: Investigation, Methodology and Validation; Z.H.: Software; Supervision. All authors have read and agreed to the published version of the manuscript.

**Funding:** This study was financially supported by National Key R&D Program of China (2018YFC1505006) and National Natural Science Foundation of China (No. 41977246).

**Data Availability Statement:** The data present in this study are openly available in the Sentinel Data Hub (https://scihub.copernicus.eu/dhus/#/home) (accessed on 25 November 2018) and DEM date repository (https://earthexplorer.usgs.gov/) (accessed on 25 November 2018). More field data and code programs can be accessible from the corresponding author upon request.

**Conflicts of Interest:** The authors declare no conflict of interest.

## Abbreviations

| Acronym | Description |
| --- | --- |
| AWEI | Automated water extraction index |
| CDAT | Change detection and thresholding |
| DEM | Digital elevation model |
| FCCs | RGB false color composites method |
| HSBA | Hierarchical split-based approach |
| HSV | Hue-saturation-value color space |
| IW | Interference wide mode |
| KI | Kittler and Illingworth |
| MNDWI | Modified normalised difference water index |
| NDVI | Normalized difference vegetation index |
| NDWI | Normalised difference water index |
| NIR | Near-infrared band |
| LiDAR | Light detection and ranging |
| OA | Overall accuracy |
| Otsu | Otsu adapting threshold |
| PA | Producer's accuracy |
| RF | Random Forest algorithm |
| SAR | Synthetic Aperture Radar |
| SRTM | Shuttle Radar Topography Mission |
| SWIR | Shortwave infrared band |
| TCT | Tasseled Cap Transformation |
| UA | User's accuracy |
| UAV | Unmanned Aerial Vehicle |
| VH | Polarization of vertical transmit-horizontal receive |
| VV | Polarization of vertical transmit-vertical receive |

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
