# Peer review of "Mapping Outburst Floods Using a Collaborative Learning Method Based on Temporally Dense Optical and SAR Data: A Case Study with the Baige Landslide Dam on the Jinsha River, Tibet"

_remotesensing, doi:10.3390/rs13112205_

Round 1

Reviewer 1 Report

In this paper, the Authors explore the benefits of suitably combining data from Sentinel-1 (SAR) and Sentinel-2 (optical) to better forecast/monitoring outburst floods. The method relies on machine learning approach (random forests) and for that, many data conversion (including colouring mapping) are done to feed the data-driven approach. 

The paper is well written, with a nice introduction. The method -not novel at all- works well for the data shown. Experimental setup is convincing (with profusion of data and using the expected metrics).

No concerns.

Author Response

Dear Reviewers:

Thank you for your letter and for the reviews’ comments concerning our manuscript entitled “Mapping outburst floods using a collaborative learning method based on temporally dense optical and SAR data: A case study with the Baige landslide dam on the Jinsha River, Tibet”(remotesensing-1230063). Your comments are valuable and helpful for revising and improving our paper, as well as the important guiding significance to our research. We have studied comments carefully and have made correction which we hope meet with approval.

Special thanks to you for your good comments.

Yours sincerely

Dr. Wei Jinbing

[email protected]

Sichuan University

Reviewer 2 Report

This paper introduced a framework by combining Sentinel-1 and Sentinel-2 satellite images for outburst flood mapping. In this regard, optical data were used to map the flooding extent in cloud-free areas using threshold analysis of different spectral indices, and SAR data were integrated with the Random Forest algorithm to map the flood extent in cloudy areas.

The method is applicable to efficiently map the flooding extent in both clear and cloudy conditions with satisfactory accuracy. The paper is concise and written well. However, several concerns should be addressed and modified before possible publication.

1- Although the authors did well with the introduction section, it would be better to include 2-3 relevant researches with brief explanations, in which Sentinel-1 and Sentinel-2 were employed in conjunction for flood mapping.

2- Please add the specification of Sentinel-1 data as separate row(s) in Table 1.

3- Line 177: Why a 7 by 7 kernel function was used? Did the authors test kernel functions with other sizes?

4- Line 196: Based on the cohesion of the content, the last sentence requires more explanation.

5- The authors stated that the required reference samples for the classification of Sentinel-1 images (in cloudy areas) were collected from Sentinel-2 images in cloud-free regions. This is a bit confusing. As far as I understand, the study area was separated into three (i.e., open flood, end flood, flood in the cloud) regions, and Sentinel-1 data were download and used over cloudy regions. So, how the training samples were transferred from cloud–free regions (in Sentinel-2) to cloudy regions to training the RF?

6- Line 209: The phrase “unknown flood pixels” is not clear for me! What does this refer to?

7- Figure 6: why is there a sudden fall and rise in NDVI and NDWI right after 0.8 km, respectively?

8- Line 323: It is recommended to include more information about Random Forest.

9- Line 348: Please clarify why this combination was used? How are these coefficients determined?

10- Comparing Figure 10(c) and Figure 11(d), the implemented approach outperformed CDAT. However, there are also some isolated pixels flooded pixels in Figure 10(c). Are these thoroughly flooded or misclassification errors?

Author Response

Dear Reviewers:

Re: Manuscript reference No. remotesensing-1230063

Thank you for your comments concerning our manuscript. I am very surprised that you read my article so carefully. Those comments are all valuable and very helpful for revising and improving our paper, as well as the important guiding significance to our research. We have studied comments carefully and have made correction which we hope meet with approval.

Please find attached a revised version of our manuscript” Mapping outburst floods using a collaborative learning method based on temporally dense optical and SAR data: A case study with the Baige landslide dam on the Jinsha River, Tibet’’, which we would like to resubmit for publication as an article. Revised portion are marked in red in the paper. The main corrections in the paper and the responds to the reviewer’s comment are as flowing:

1 Although the authors did well with the introduction section, it would be better to include 2-3 relevant researches with brief explanations, in which Sentinel-1 and Sentinel-2 were employed in conjunction for flood mapping.

Response: We have added 5 related articles to enrich the introduction in this part. A brief explanation is given on the advantages of conjunction uses with Sentinel-1 and Sentinel-2 date. We summarize the main combination modes of Sentinel-1 and Sentinel-2 date in the end. These changes following the comments do help improve the article.

The additions are as follows (Line 90-96):

“A number of studies have demonstrated the potential of combining Sentinel-1 data with Sentinel-2 images to support multi-scale flood damage assessment [32], automated training data selection [33] and an increase in observation density during flooding events [34]. The existing combinatorial modes of optical, SAR data for flood mapping usually put the Sentinel-2 multispectral optical image as the central part. VV / VH polarization’s backscattering coefficient obtained by processing Sentinel-1 SAR images is added as a separate variable [35,36].”

2- Please add the specification of Sentinel-1 data as separate row(s) in Table 1.

Response: The specification of Sentinel-1 image parameters is added as a separate Table 2. (Line 194)

Acquisition time before flood

20181103T23:19

Polarization

VV/VH

Acquisition time after flood

20181115T23:20

Incidence angle

39.56°

Number of SAR scenes

6

Product type

IW-GRD

Spatial resolution

10 m

View geometry

Descending

wavelength

5.6 cm

Orbit cycle

154

Table 2. Sentinel-2 optical images characteristics used for this study.

3  Line 177: Why a 7 by 7 kernel function was used? Did the authors test kernel functions with other sizes?

Response: The authors did not test kernel functions with other size. Refined Lee filter with a window size of 7 × 7 pixels is default parameter in the software SNAP 8.0. The size has been generally used in other studies with pre-processing of Sentinel-1 dates, therefore, it should be appropriate.

 Relevant references are as follows:

(1) Slagter, B., Tsendbazar, N., Vollrath, A. and Reiche, J., 2020. Mapping wetland characteristics using temporally dense Sentinel-1 and Sentinel-2 data: A case study in the St. Lucia wetlands, South Africa. International Journal of Applied Earth Observation and Geoinformation, 86, p.102009. https://doi.org/10.1016/j.jag.2019.102009

(2) Whyte, A., Ferentinos, K. and Petropoulos, G., 2018. A new synergistic approach for monitoring wetlands using Sentinels -1 and 2 data with object-based machine learning algorithms. Environmental Modelling & Software, 104, pp.40-54. https://doi.org/10.1016/j.envsoft.2018.01.023.

(3) Lee, J., Ainsworth, T., Wang, Y. and Chen, K., 2015. Polarimetric SAR Speckle Filtering and the Extended Sigma Filter. IEEE Transactions on Geoscience and Remote Sensing, 53(3), pp.1150-1160. https://doi.org/10.1109/tgrs.2014.2335114.

4 Line 196: Based on the cohesion of the content, the last sentence requires more explanation.

Response: we made more explanation for the last sentence.

The sentence rewrote as follows (Line 211):

“The cloud-covered area accounted for 176.66 km2 and distributed on four sections (C1, C2, C3, and C4)   in ongoing flood and end-of-flood regions, which occupied 30.08 % of the total work area.”

5 The authors stated that the required reference samples for the classification of Sentinel-1 images (in cloudy areas) were collected from Sentinel-2 images in cloud-free regions. This is a bit confusing. As far as I understand, the study area was separated into three (i.e., open flood, end flood, flood in the cloud) regions, and Sentinel-1 data were download and used over cloudy regions. So, how the training samples were transferred from cloud–free regions (in Sentinel-2) to cloudy regions to training the RF?

Response: The study area was separated into three (i.e., open flood, end flood, flood in the cloud) regions, and Sentinel-1 data were download and used over cloudy regions. The scope of Sentinel-1 data was more than cloudy regions. In the cloudless region, we completed the training process and obtained the optimal prediction parameters for the RF model. Sentinel-1 SAR images and DEM in those cloudless areas provided relevant features and the flood pixels from Sentinel-2 images provided a label set. In cloudy regions, based on the features set from Sentinel-1 SAR images and DEM data and prediction parameters, we predicted the flood range.

6- Line 209: The phrase “unknown flood pixels” is not clear for me! What does this refer to?

Response: it is really true as Reviewer suggested that this sentence here was unclear. We changed the “unknown flood pixels” to “the pixels of unknown category”.

The sentence rewrote as follows (Line 227):

“Then, the flood pixels obtained from the Sentinel-2 optical images in the cloudless area served as the label set of the training samples, and the pixels of unknown category in the cloud-covered area were used as the prediction samples for the output.”

7 Figure 6: why is there a sudden fall and rise in NDVI and NDWI right after 0.8 km, respectively.

Response: Near November 19, the region experienced rainfall, resulting in a sudden change in value. As a result, the ground moisture after 0.8 km in non-flooded areas is higher than that of clear day before. Therefore, the overall value of NDVI curve on November 19 was lower than that on November 14, showing a decrease. The NDWI curve was higher than that on November 14, showing an increase.

8 Line 323: It is recommended to include more information about Random Forest

Response: It is a good suggestion to introduce the main algorithms of our paper in more detail. The introduction of essential features and operating principle about RF algorithm is supplemented in the paper.

The following are added (Line 343-348):

“The RF model is composed of many individual decision tree models that each tree fits a data subset sampled independently using bootstrapping. Out of Bag data is used to get both variable importance estimations and an internal unbiased classification error as trees are added to the forest, while bagging form of bootstrapping is used to randomly select samples of variable as the training dataset for model calibration [47].”

9- Line 348: Please clarify why this combination was used? How are these coefficients determined?

Response: Sentinel-1 collects images in VH and VV polarization, both of which have the potential superiority for flood classification. VV polarization has an advantage in distinguishing surface roughness changes, while the VH polarization is more sensitive to scattering changes of specific ground objects, such as vegetated land. Previous research concluded that VV provides a slight advantage when using Sentinel-1 data for flood mapping. Both of VH and VV polarization have the potential to generate for classification errors and the limitation of each polarization as environmental conditions vary requires acknowledgement. To allow for fully extracting the change information of the backscatter coefficient, this combination and intersect are necessary.

   We downloaded a series of Sentinel-1 data freely at the Copernicus open access hub from European Space Agency (https://scihub.copernicus.eu/dhus/#/home). Major pre-processing procedures for Sentinel-1 were executed on SNAP 8.0.0 (http://step.esa.int/main/download/snap-download/). The following processing steps include orbit correction, thermal noise Removal, border noise removal, radiometric calibration, multi-looking, terrain correction speckle filtering. Finally, we convert linear data to decibel (dB) and generated the backscatter coefficient of VV/VH.

   On November 15, 2018, the VV/VH-polarized SAR image, was taken as the master images, and made the difference or ratio with the image on November 03, 2018, to get the backscatter coefficient change map. Since there are only 13 variables in total, they are not filtered and all are input into the machine learning model.

10 Comparing Figure 10(c) and Figure 11(d), the implemented approach outperformed CDAT. However, there are also some isolated pixels flooded pixels in Figure 10(c). Are these thoroughly flooded or misclassification errors?

Response: There are also some isolated pixels flooded pixels in Figure 10(c) and these are misclassification errors. Compared to the result of CDAT, both flood pixels and non-flood pixels have been improved notably, showing a better monitoring ability than existing approaches. However, the current classification method is still based on pixels, and an object-oriented machine learning classification method may have a better classification result. In addition, we need to continue to explore advanced algorithms. In some application scenarios, deep learning algorithms with convolutional neural networks have better re-recognition and classification performance than other machine learning models.

We shall look forward to hearing from you at your earliest convenience.

Yours sincerely

Dr. Wei Jinbing

[email protected]

Sichuan University

Reviewer 3 Report

Dear authors,

There are some minor mistakes on English grammar, please revise:

Abstract, Line 17: delete "in recent year". Rewrite this sentence as bellow: 
There have been significant advances in flood mapping using remote sensing images in recent years, but little attention has been devoted to outburst flood mapping.

Line 43: The word serious is often overused. change to severe.

page 2, Line 56: it seems that "outburst" may not agree in number with the other words in this phrase. 

Page 2m Line 57: the word "difficult" is often overused. Consider using a more specific synonym to improve the sharpness of your writing, change to "challenging"

Page 2, Line 58: the word "common" is often overused. Consider to change "standard, or expected"

Page 2, Line 62: The subordinate phrase: "Owing to using different... and receiving radar signals" does not appear to be modifying the subject "a radar system". please rewrite the sentence to avoid a dangling modifier.

Page 2: Line 83: Your sentence may be unclear or hard to follow. Consider delete "between", delete "been seldom', change to "has seldom been researched.

Page 2, Line 88: this sentence may be unclear, please delete "it has been proved its', change to "Its good performance in the classification has been proved with a mass of satellite image data [31]

Page 2, Line 93: your sentence may be hard to follow. Please delete "applications of", rewrite: While both optical and SAR images have been widely used for open water delineation and flood mapping, there have been few short-duration explosive flood mapping applications.

The other parts are good writing.

There is no duplicate in this study.

Thanks.

Author Response

Dear Reviewers:

Re: Manuscript reference No. remotesensing-1230063

Your comments were highly insightful and enabled us to greatly improve the quality of our manuscript. In the following pages are our point-by-point responses to each of the comments of your own comments.

Please find attached a revised version of our manuscript” Mapping outburst floods using a collaborative learning method based on temporally dense optical and SAR data: A case study with the Baige landslide dam on the Jinsha River, Tibet’’, which we would like to resubmit for publication as an article. Revised portion are marked in red in the paper. The main corrections in the paper and the responds to the reviewer’s comment are as flowing:

1 Abstract, Line 17: delete "in recent year". Rewrite this sentence as bellow: “There have been significant advances in flood mapping using remote sensing images in recent years, but little attention has been devoted to outburst flood mapping.”

   Response: We rewrote this sentence as bellow (Line 17):

“There have been significant advances in flood mapping using remote sensing images in recent years, but little attention has been devoted to outburst flood mapping.”

2 Line 43: The word serious is often overused. change to severe.

Response: We changed the word” serious” with “severe”

3 page 2, Line 56: it seems that "outburst" may not agree in number with the other words in this phrase.

Response: We changed “outburst flood “as “the outburst flood”, which was agree in number with the other words.

4 Page 2 Line 57: the word "difficult" is often overused. Consider using a more specific synonym to improve the sharpness of your writing, change to "challenging".

Response: We used a more specific synonym "challenging" instead of "difficult".

5 Page 2, Line 58: the word "common" is often overused. Consider to change "standard, or expected".

Response: We replaced the word "common" with the word “expected” to make the expression more accurate.

6 Page 2, Line 62: The subordinate phrase: "Owing to using different... and receiving radar signals" does not appear to be modifying the subject "a radar system". please rewrite the sentence to avoid a dangling modifier.

Response: We rearranged the order of the languages to avoid a dangling modifier.

The sentence was rewrite as followed (Line 63):

“Owing to transmitting and receiving radar signals in different polarization modes, a radar image can obtain a rich polarization scattering matrix to reflect the inherent characteristics of ground objects.”

7 Page 2: Line 83: Your sentence may be unclear or hard to follow. Consider delete "between", delete "been seldom', change to "has seldom been researched.

Response: We deleted "between" and change "been seldom” to "has seldom been researched. The sentence was rewrite as followed (Line 89):

The combined use of time-intensive Sentinel-1 and Sentinel-2 data to improve classification accuracy and timeliness has seldom been researched.”

8 Page 2, Line 88: this sentence may be unclear, please delete "it has been proved its', change to "Its good performance in the classification has been proved with a mass of satellite image data.

Response: We rewrote this sentence to make it clear and concise.

The sentence was rewrite as followed (Line100):

“Its good performance in the classification has been proved with a mass of satellite images data.”

9 Page 2, Line 93: your sentence may be hard to follow. Please delete "applications of", rewrite: While both optical and SAR images have been widely used for open water delineation and flood mapping, there have been few short-duration explosive flood mapping applications.

Response: We changed the order of the word " applications " in the sentence to make it clearer. The sentence was rewrite as followed (Line 105):

While both optical and SAR images have been widely used for open water delineation and flood mapping, there have been few short-duration explosive flood mapping applications.

We shall look forward to hearing from you at your earliest convenience.

Yours sincerely

Dr. Wei Jinbing

[email protected]

Sichuan University

Reviewer 4 Report

A BRIEF SUMMARY

The paper titled “Mapping outburst floods using a collaborative learning method based on temporally dense optical and SAR data: A case study with the Baige landslide dam on the Jinsha River, Tibet” presents a good topic for readers of this Journal. The topic represents an interesting line of research. The paper is very well structured. Results are good described and analysed in the paper. I have only some minor comments.

At lines 79-81: You have to add more information on dem-based flooding evaluation. In particular, to improve introduction, you have to consider some recent studies on this topic. For example:

https://doi.org/10.3390/w12061717

https://doi.org/10.1038/sdata.2018.309.

Figure 2: I do not understand where localized pictures a-b-c on the reported map.  

Figure 7: What did you represent with letters (D-B-F-A-C)? Can you explain, please?

Author Response

Dear Reviewers:

Re: Manuscript reference No. remotesensing-1230063

Please find attached a revised version of our manuscript” Mapping outburst floods using a collaborative learning method based on temporally dense optical and SAR data: A case study with the Baige landslide dam on the Jinsha River, Tibet’’, which we would like to resubmit for publication as an article.

I have read the recent studies on this topic you recommend carefully. These researches are very valuable and helpful to improve my paper. Thank you very much for your efforts to improve our manuscript.

Responses to the comments of Reviewer #4

1 At lines 79-81: You have to add more information on dem-based flooding evaluation. In particular, to improve introduction, you have to consider some recent studies on this topic. For example:

https://doi.org/10.3390/w12061717

https://doi.org/10.1038/sdata.2018.309.

Response: we added more information on dem-based flooding evaluation to improve introduction.

After reading some recent studies on this topic, the latest research is added as follows(Line 82-87):

Nardi et al [28] present the first global floodplain dataset at 8.33 arcsecond resolution with the Shuttle Radar Topography Mission (SRTM) digital terrain model. Some emerging technologies, such as Unmanned Aerial Vehicle (UAV) and Light Detection and Ranging (LiDAR), can provide high resolution and accurate DEMs to support flood mapping and depth simulation in small-scale [29].

2 Figure 2: I do not understand where localized pictures a-b-c on the reported map.

Response: We are very sorry for our negligence of the location of pictures a-b-c on Figure 2. The new Figure 2 indicating the location of pictures (a)-(b)-(c) replaced the previous figure. The location described in the text with word as well, such as “(Fig. 2a, Zhu balong country, NO 17)”, “(Fig. 2b, Ya lang country, NO 23)” and “(Fig. 2c, Ta chen country, NO 36)”

3 Figure 7: What did you represent with letters (D-B-F-A-C)? Can you explain, please?

Response: The letters (A-B-C) represent the feature points on the boundary between actual flood and non-flood zones. The shape of flood boundary in flood mapping is consistent with that of real flood photo. The location of the convex point A-B-C in the flood boundary is consistent. Meanwhile, the letters (D-F) represent the feature points of non-submerged area. The highland F in the left bottom region and the lowland in the left corner D are remarkably consistent with the actual not-flooded range. To make the content clearer, we use different colors represent feature points with different meanings in the new Figure 7.

We shall look forward to hearing from you at your earliest convenience.

Yours sincerely

Dr. Wei Jinbing

[email protected]

Sichuan University

Round 2

Reviewer 2 Report

The authors have answered/applied my comments properly.